# Review of the National Program for Onchocerciasis Control in the Democratic Republic of the Congo

**DOI:** 10.3390/tropicalmed4020092

**Published:** 2019-06-13

**Authors:** Jean-Claude Makenga Bof, Fortunat Ntumba Tshitoka, Daniel Muteba, Paul Mansiangi, Yves Coppieters

**Affiliations:** 1Ecole de Santé Publique, Université Libre de Bruxelles (ULB), Route de Lennik 808, 1070 Brussels, Belgium; yves.coppieters@ulb.ac.be; 2Ministry of Health: Program of Neglected Tropical Diseases (NTD) for Preventive Chemotherapy (PC), Gombe, Kinshasa, DRC; fntumbaliz@gmail.com (F.N.T.); danielmuteba2002@yahoo.fr (D.M.); 3Faculty of Medicine, School of Public Health, University of Kinshasa (UNIKIN), Lemba, Kinshasa, DRC; pmansiangi@gmail.com

**Keywords:** onchocerciasis, ivermectin, review, the DRC, background, national program

## Abstract

Here, we review all data available at the Ministry of Public Health in order to describe the history of the National Program for Onchocerciasis Control (NPOC) in the Democratic Republic of the Congo (DRC). Discovered in 1903, the disease is endemic in all provinces. Ivermectin was introduced in 1987 as clinical treatment, then as mass treatment in 1989. Created in 1996, the NPOC is based on community-directed treatment with ivermectin (CDTI). In 1999, rapid epidemiological mapping for onchocerciasis surveys were launched to determine the mass treatment areas called “CDTI Projects”. CDTI started in 2001 and certain projects were stopped in 2005 following the occurrence of serious adverse events. Surveys coupled with rapid assessment procedures for loiasis and onchocerciasis rapid epidemiological assessment were launched to identify the areas of treatment for onchocerciasis and loiasis. In 2006, CDTI began again until closure of the activities of African Program for Onchocerciasis Control (APOC) in 2015. In 2016, the National Program for Neglected Tropical Diseases Control using Preventive Chemotherapy (PNMTN-CP) was launched to replace NPOC. Onchocerciasis and CDTI are little known by the population. The objective of eliminating onchocerciasis by 2025 will not be achieved due to the poor results of the NPOC. The reform of strategies for eliminating this disease is strongly recommended.

## 1. Introduction

Onchocerciasis, also known as river blindness, volvulosis, or Robles disease, is a cutaneous filariasis [1], caused by the filarial nematode *Onchocerca volvulus* [2]. The worm infests the skin and eyes and is transmitted from an infected individual to a healthy individual via the bite of an infected black fly, belonging to the Simuliidae family [3]. The principal manifestations are cutaneous (pruritis, filarial itch, subcutaneous nodule, lizard skin, leopard skin, etc.) and ocular (keratitis, uveitis, atrophy of the optic nerve, chorioretinitis, etc.), with blindness being the most serious complication [4]. Onchocerciasis represents the second most common cause of blindness of infectious origin in the world after trachoma [5]. According to the World Health Organization (WHO) in 2018, approximately 120 million people worldwide are exposed to the risk of onchocerciasis, 96% of whom are in Africa [6]. The 38 countries where onchocerciasis is endemic include, on the one hand, 31 countries in sub-Saharan Africa, including Angola, Benin, Burkina Faso, Burundi, Cameroon, Ivory Coast, Ethiopia, Gabon, Ghana, Guinea, Guinea Bissau, Equatorial Guinea, Kenya, Liberia, Malawi, Mali, Mozambique, Niger, Nigeria, Uganda, Central African Republic, the Democratic Republic of the Congo, United Republic of Tanzania, Rwanda, Senegal, Sierra Leone, Sudan, Southern Sudan, Chad, and Togo. On the other hand, the other seven countries are Yemen in Asia and six countries of North and South America (Brazil, Colombia, Ecuador, Guatemala, Mexico, and Venezuela) [7].

Worldwide, since 2016, it is estimated that there are 18 million people who are infected and present dermal microfilaria; 99% of these people are in Africa [6]. The Global Burden of Disease Study estimated in 2017 that there were 20.9 million prevalent *O. volvulus* infections worldwide; 14.6 million of the infected people had skin disease and 1.15 million had vision loss [7]. This high prevalence shows that onchocerciasis constitutes an urgent public health problem in the countries mentioned, generally accompanied by serious socio-economic repercussions [1,6,7]. The first confirmed elimination of an onchocerciasis focus in Africa occurred in Abu Hamed, Sudan, in 2016 [8]. Still in 2016, Guatemala became the fourth country in the world to be declared free of onchocerciasis, after Colombia in 2013, Ecuador in 2014, and Mexico in 2015, after the successful application of elimination activities for decades [7]. By the end of 2017, three additional countries stopped mass drug administration and completed three years of post-treatment surveillance in at least one transmission area: Bolivarian Republic of Venezuela, Uganda, and Sudan [7].

In the Democratic Republic of the Congo (DRC), the disease is known since 1903, thanks to the work of Brumpt along the Uele River [9]. Since 2016, 38 million people, some 41% of the Congolese population, are believed to be at risk of contracting onchocerciasis, and 65 thousand people (1‰ of the population) suffer from blindness [10,11]. Prior to the creation of the National Program for Onchocerciasis Control (NPOC) in the DRC, the World Bank realized via a study that this disease caused affected persons to lose several years of working life [12]. Furthermore, WHO noted that carriers of the disease are isolated and socially abandoned [1]. The NPOC claims that children who are obliged to act as guides for their parents cannot continue their education [13].

The DRC is a country with particularly rich natural resources and raw materials. However, the population seems to be among the poorest in the world. In effect, life expectancy is estimated at 52 years and the gross domestic product (GDP) amounts to just 3.3 United States dollars (USD) per inhabitant; thus, the presence of onchocerciasis plays a part in the fall in economic forces and defies the efforts of the international community to control poverty [14].

To control this disease, several approaches were tested: (i) treatment with diethylcarbamazine (DEC) used clinically and in mass treatment, later abandoned due to its side effects, which were often serious and intolerable in the treated communities; (ii) vector control, by spreading insecticides: dichloro diphenyl trichloroethane (DDT) and temephos (Abate^®^) carried out in 1967, particularly in the Inga and Lusambo foci, with relatively satisfactory results (however, due to the very high cost, it was stopped in 1975); (iii) treatment with ivermectin, introduced in 1987 by the Non-Governmental Development Organization (NGDO) initially in the form of clinical treatment in the Kasaï and Uele foci, before being used in 1989 as mass treatment [15]. The NPOC, whose strategy is based on community-directed treatment with ivermectin (CDTI), was created in 1996. Thanks to this program, the DRC was able to start the rapid epidemiological mapping of onchocerciasis (REMO) surveys in 1999 to determine the mass treatment areas known as “CDTI projects” [13,16]. REMO is a rapid assessment of onchocerciasis endemicity by nodule detection. In the DRC, at least 30 villages are selected, including 30 to 50 people aged 20 years and over per village—who lived more than 10 years in the village—with a distance of 30–50 km between two villages along the main river. If ≥20% of adults have nodules, mass treatment is required, and this figure is extrapolated to the entire area. In communities where the nodule rate is less than 20%, clinical treatment is applied [13].

CDTI began in 2001 and was suspended in 2005 in certain projects due to the occurrence of deaths caused by the population taking ivermectin in health zones where onchocerciasis and loiasis occurred together. Consequently, the combined use of rapid assessment procedures for loiasis (RAPLOA) and onchocerciasis rapid epidemiological assessment (REA) surveys was launched to precisely identify the areas of treatment for onchocerciasis and to separate the areas where the two filariases exist in co-endemicity [13,16]. REA is a technique that allows collecting, quickly and at a lower cost, various information related to the symptoms of onchocerciasis. It is a revised form of REMO [13]. CDTI resumed in 2006 until the end of the activities of the African Program for Onchocerciasis Control (APOC) in 2015. At that point, the Congolese government then took charge of the continuation of activities.

Therapeutic coverage denotes the number of people treated multiplied by 100 and divided by the total population (exposed population). It must be equal to or greater than 84% each year in each community. Geographical coverage, on the other hand, denotes the number of communities treated multiplied by 100 and divided by the total number of communities. It must reach 100% each year [13]. Although APOC initially recommended a minimum therapeutic coverage of ≥65% for control of onchocerciasis as a public health problem, when its strategy moved from control toward elimination, this target was elevated to ≥80% with a recommended geographical coverage of 100% [13]. Therapeutic coverage higher than or equal to 80% and a geographic coverage of 80% to 100% are required for at least 15 to 17 annual cycles to conquer this scourge [10]. Unfortunately, the DRC’s NPOC did not reach either of these goals between the years of CDTI control from 2001 to 2012 [17].

Currently, the geographic coverage of the treatment is not complete because CDTI is not integrated in some health zones where the co-endemicity of onchocerciasis and loiasis is an obstacle to the administration of ivermectin. The reports received from the NPOC concerning the distribution of ivermectin over the last 15 years show that average therapeutic and geographic coverages reached 74.1% and 63.3%, respectively [10]. These results are not very reassuring with regard to elimination of the disease. It must be recognized that the extent of therapeutic and geographic coverage originates from the coordination of activities within the NPOC. As the methods for onchocerciasis control in the DRC are not well documented, the aim of our study was to describe the history of the NPOC. It is, therefore, a historical review of the programs for onchocerciasis control on a worldwide scale and in particular in DRC. This review will make it possible to understand the current status of the situation and plan for future challenges in order to fight this disease better in the DRC.

## 2. Methodology

This is a documentary, descriptive, and analytic review of the data available at the Ministry of Public Health in DRC, which was carried out from 1 September 2017 to 31 August 2018. The main documents consulted were the following: (i) documents on the policy of controlling onchocerciasis in DRC and the operational strategies of NPOC; (ii) legal texts mentioning the regulation for onchocerciasis control in the DRC; (iii) data regarding the distribution of ivermectin, namely the number of projects implemented, the total population of the meso/hyperendemic zones, the number of villages in the meso/hyperendemic zones, the total population of the meso/hyperendemic zones (having accepted, refused, or abandoned treatment), the number of health workers trained, the number of community-based distributors (CD) trained, geographic coverage, therapeutic coverage, and the number of tablets distributed, expired, and unused; (iv) documents from the projects and other interventions developed by the partner institutions of the Ministry of Health concerning onchocerciasis control in the DRC; (iv) data on the financing of CDTI activities; (v) annual technical reports from CDTI projects from 2000 to 2016 compiled at the NPOC level; (vi) annual reports of the meetings of the WHO Technical Advisory Committee and of the African Program for Onchocerciasis Control (APOC), as well as the training and supervision reports drawn up by the CDTI projects; (vii) reviews of the growth and poverty reduction strategy document of the DRC: health thematic report, Kinshasa 2000–2018; (viii) published scientific articles; (ix) WHO weekly epidemiological records; and (x) websites of various departments of the Congolese government connected with onchocerciasis.

To better tell the story of the NPOC, we conducted our documentary review in the following manner: firstly, on the basis of the framework described by Durocher et al., we examined all the scientific articles published in connection with onchocerciasis in the DRC [18]. Secondly, we selected and examined all the reports, legal texts, and databases available at the Ministry of Public Health or elsewhere (APOC, WHO) and called upon the experts from the NPOC for further clarification on certain points. Thirdly, we consulted the archives of the Congolese national journals and the weekly epidemiological records of the WHO. Finally, we visited some websites connected with onchocerciasis in the DRC.

The criteria for exclusion and inclusion of all the documents, databases, and websites were deliberately flexible. Relevant unpublished documents were also consulted and taken into account for this review.

## 3. Results

### 3.1. History of the Programs for Onchocerciasis Control in the World 

Several programs were created to control the disease. The most well-known of these include the Onchocerciasis Control Program in West Africa (OCP), launched in 1974 in collaboration with four agencies of the United Nations: the WHO, the World Bank, the United Nations Development Program (UNDP), and the Food and Agriculture Organization (FAO). These agencies also provided the financing for the OCP. This program stretched over 1,200,000 km² to protect 30 million people from the consequences of “river blindness” [19,20]. The countries covered by the OCP were Benin, Burkina Faso, Ivory Coast, Ghana, Guinea Bissau, Guinea, Mali, Niger, Senegal, Sierra Leone, and Togo. The estimated total cost of the program was 550 million USD, less than 1 USD per year for each person protected. The efforts of the OCP allowed the disease to be controlled in West Africa between 1974 and 2002 [19,20] (Figure 1).

For 14 years, OCP operations were based exclusively on the spreading of insecticides by helicopters and aircraft over the breeding sites of black flies in order to kill their larvae. The announcement of the donation of Mectizan^®^ (ivermectin) by Merck & Co. Inc. in 1987 made it possible in 1989 to add ivermectin treatment to exclusive vector control by larvicides. The OCP was officially closed in 2002 after stopping transmission of the disease in all the participating countries apart from Sierra Leone, where operations were interrupted by a 10-year civil war [19,20]. The global benefit of this OCP operation was to prevent almost 600 thousand cases of blindness, to save 18 million children born in the areas now controlled in terms of the risk of “river blindness”, and to bring 25 million hectares of land under cultivation. OCP clearly demonstrated the vital role played by this partnership in the improvement of health and its impact on socio-economic development in remote and neglected areas [19] (Figure 1).

The Onchocerciasis Elimination Program for the Americas (OEPA) is the second program, created in 1992 with the support of the Pan American Health Organization (PAHO), the Inter-American Development Bank, a consortium of NGDOs, and six countries of North and South America (Brazil, Columbia, Ecuador, Guatemala, Mexico, and Venezuela) in order to coordinate efforts to control onchocerciasis in these endemic countries, to provide care for and eliminate the disease [1] (Figure 1).

The remarkable success of the OCP from the point of view of health, the economy, and development served as justification for the launch of a third program in 1995, the African Program for Onchocerciasis Control (APOC). The sponsoring organizations and donors were the same as for the OCP. However, unlike the OCP, this new program is not vertical (specialized) and rests on a full partnership between the affected communities, participating governments, a consortium of NGDOs, and bilateral organizations. The aim of this program is to set up, over a period of 12 years, durable systems for the community-directed distribution of ivermectin (Mectizan^®^) and cover some 50 million people in 19 countries which are not part of the OCP and in which onchocerciasis remains a serious public health problem. The following countries are covered: Angola, Burundi, Cameroon, Congo, Ethiopia, Gabon, Equatorial Guinea, Kenya, Liberia, Malawi, Nigeria, Uganda, Central African Republic, the Democratic Republic of the Congo, Rwanda, Sudan, Tanzania, and Chad [1]. The APOC partners are jointly responsible for implementation of the program’s main control strategy, which is based on CDTI. Where feasible, ivermectin treatment will be supplemented via elimination of vectors using environmentally safe methods [1] (Figure 1).

In 1996, the NGDO coordination group supervised the distribution of ivermectin to 7.5 million people. During the first year of the APOC’s activities in the field (1997–1998), this number rose to 11.7 million, and would exceed 15 million in 1999. This constant increase, together with the close working relationship between all its partners, allowed the APOC to achieve its objective of treating 45 million people in 2007 [1] (Figure 1).

When the OCP stopped its activities in 2002, the strategy of eliminating larvae via aerial spraying was no longer applied, while the distribution of ivermectin became the sole method of control in some countries covered by the OCP. These countries continued ivermectin treatment using the same control strategy as the APOC and the challenges were the same, listed below.

(1)Development of competent distribution systems with the affected communities, which served as an example for the distribution of other medicines for treating neglected tropical disease.(2)At the end of the activities of the OCP, this program aimed to integrate the delegated activities into the various health systems, whereas the aim of the APOC from the outset was to delegate all control activities to the health systems of the participating countries. In this area, these two programs exchanged their experience.(3)Moreover, the additional challenge for APOC was to show that its partnership was capable of providing a lasting solution to a public health and development problem [1] (Figure 1).

In summary, Figure 1 highlights the Onchocerciasis Elimination Program for the Americas (OEPA) created in 1992 in America, the Onchocerciasis Control Program (OCP) established between 1974–2002 in West Africa, and the African Program for Onchocerciasis Control (APOC) founded in 1995 in all countries in Africa (Figure 1). It should be noted that the Yemeni Elimination Program for Onchocerciasis (YEPO) was created in 2001 in Yemen (Asia).

### 3.2. History of the Programs for Onchocerciasis Control in DRC

In DRC, onchocerciasis control was provided by the NPOC, created in 1996 by the Ministry of Health, until 2016. In 1999, via the technical support of WHO/APOC, the NPOC established the areas of mass treatment known as “CDTI projects” based on REMO (rapid epidemiological mapping of onchocerciasis) surveys [17] (Figure 2). Levels of onchocerciasis endemicity were defined as follows: sporadic zone (prevalence of nodules <10%), hypoendemic zone (10–19.9%), mesoendemic zone (20–39.9%), and hyperendemic zone (≥40%). If more than 20% of adults had nodules, mass treatment was necessary, and this figure was extrapolated to the zone as a whole. In communities where the level of nodules was less than 20%, treatment was administered via clinics [17] (Figure 2).

Before the transition from control to elimination of onchocerciasis, the DRC included only in treatment projects (hotbed) with a nodular prevalence of 20% or more. All Health Zones (HZ) with a nodular prevalence of less than 20% were considered hypoendemic and, therefore, not eligible for treatment. With the objective of eliminating onchocerciasis, the WHO currently recommends that elimination mapping should be redone in areas of unknown status (i.e., hypoendemic areas at the time) and in non-endemic areas. Therefore, all endemic areas must be treated [13] (Figure 2).

In 2016, the NPOC (1996–2015) was relayed by the National Program for the Control of Neglected Tropical Diseases through Preventive Chemotherapy (NPCNTDs-PC), whose mission was to eliminate “neglected tropical diseases amongst which onchocerciasis”, and to reduce its consequences on socio-economic development. The control strategies concerning onchocerciasis were, firstly, CDTI and, secondly, anti-vectorial measures [11]. Vector control is currently applied only in one hotbed, North Ituri, where traps are used to catch black flies. It should be noted that, with the establishment of the committee of independent experts on onchocerciasis control, this control strategy is a priority to achieve the elimination of onchocerciasis in the DRC [13]. In contrast, the strategies recommended by WHO to control neglected tropical diseases (NTDs) are chemoprevention, anti-vectorial measures, intensified care of cases, environmental sanitation, and veterinary health [23].

The integration of rapid cartography of onchocerciasis and loiasis is ongoing in the country (Figure 3). This cartography not only reduces the time and cost of surveys, but also facilitates operational decision-making regarding ivermectin treatment in zones where loiasis may be co-endemic [24] (Figure 3). One recent approach suggests the need to use even finer than normal high-resolution cartography before implementing activities to control lymphatic filiariasis. The use of “micro-stratification overlap mapping” (MOM) is essential for planning the widespread distribution of medicines for lymphatic filiariasis programs in countries with co-endemic filarial infections [25].

NPCNTDs-PC functions with normative documents such as the national health policy and sectoral strategic plans. Some of these norms are not adopted by the government. Furthermore, the policies and sub-sectoral plans are not always in step with the national health development plan based on primary healthcare [11].

### 3.3. Success of NPOC in Endemic Areas Other than the DRC

After having carefully implemented the control strategies recommended by the APOC, onchocerciasis control programs in several countries were successful, going from control to complete elimination of the disease: Colombia in 2013, Ecuador in 2014, Mexico in 2015, and Guatemala and Sudan in 2016 [7,8].

In addition, toward the end of 2017, other countries, such as the Bolivarian Republic of Venezuela, Uganda, and Sudan, also terminated CDTI in at least one transmission zone, conducted for three years post-treatment surveillance [7].

In general, the elimination of onchocerciasis is possible after strict application of control strategies as recommended by WHO. Each country has its own particularities in order to adapt these strategies according to the reality of their own situation.

We, therefore, encourage the NPCNTDs-PC to develop efforts and multiply control strategies to eliminate onchocerciasis. Indeed, the results of NPOC, such as the increase in therapeutic and geographical coverage from year to year and the increase in the number of community distributors (CDs) and trained health workers (HWs), are to be encouraged. In the first 12 years of CDTI in the DRC, the country achieved an average of one CD per 262 people and one HW per 2368 people, while the standards recommended by the APOC are one CD per 100 people and one HW per 2500 to 5000 people [17].

Difficulties at the beginning included armed conflicts (1998–2003–2007), serious side effects (death), and the objectives of therapeutic and geographical coverage not achieved due to geographical inaccessibility.

### 3.4. Global Control Strategy

Developed during the 1980s, ivermectin is the first medicine capable of effectively and safely reducing the number of cutaneous microfilariae present in onchocerciasis patients and guaranteeing clinical improvement and reduction in transmission. It was, thus, possible to define a new global strategy to control the disease based on the annual administration of a single dose of ivermectin to affected populations [19,20]. In 1987, the producer of ivermectin, Merck & Co. Inc., undertook to provide, free of charge and for as long as necessary, the quantities of medicines needed to eliminate onchocerciasis as a public health problem. In collaboration with WHO, the Ministries of Health, and the NGDOs, it set up a Mectizan^®^ donation program. Thus, between 1987 and 1996, more than 65 million doses of Mectizan^®^ were distributed free of charge [19,20].

In the DRC, since 2001, mass treatment with ivermectin started with the CDTI Kasaï project. Other CDTI projects were integrated progressively until almost 90% of meso- and hyperendemic villages in the country were covered. However, therapeutic coverage of 80% or more and a geographic coverage of 80% to 100% are required for at least 15 to 17 annual cycles to eliminate onchocerciasis [10]. Unfortunately, in 15 years of CDTI, the DRC never reached these recommended guidelines in terms of geographical and therapeutic coverage.

From 2004 to 2005, however, the Congo Central/Kinshasa, Tshopo, and Uele projects, as well as other projects launched a few months earlier, were suspended temporarily following deaths associated with the use of ivermectin in populations where onchocerciasis and loiasis coexisted. For this reason, the combined use of mapping of loiasis using the rapid assessment for loiasis (RAPLOA) method and of onchocerciasis using rapid epidemiological assessment (REA) was started to identify treatment zones for onchocerciasis and to exclude zones of hyperendemic loiasis. In 2005, only the populations of the Bandundu, Kasaï, and Sankuru projects were treated with ivermectin. In 2006, community-directed treatment with ivermectin was organized in the Bandundu, Congo Central/Kinshasa, Équateur-Kiri, Kasaï, Katanga Nord and Katanga Sud, Lualaba, Mongala, Rutshuru-Goma (Nord-Kivu province), Sankuru, Tshopo, Tshuapa, Ubangi du Nord, Sud Ubangi, and Uele projects. From 2007 to 2015, treatment continued year after year with much difficulty. On the one hand, insecurity, geographical inaccessibility, and serious adverse effects did not allow treatment to be conducted normally, and, on the other hand, information, awareness, and education sessions were not correctly organized to enable the different communities to understand the mass treatment and to participate widely [10]. In 2015, after APOC ceased its activities, CDTI continued under the control of the Ministry of Public Health of DRC via the Onchocerciasis Elimination Project created by WHO in the same year and by the Independent Committee for Monitoring and Eliminating Onchocerciasis [13].

### 3.5. Epidemiology and Geographical Distribution of Onchocerciasis in DRC

In the DRC, mapping of onchocerciasis using the REMO (rapid epidemiological mapping of onchocerciasis) method reveals that the disease is present in all 26 provinces of the country, at various levels of endemicity (Figure 4). The prevalence of nodules varies from 1% to 100% [11,12]. For this reason, onchocerciasis represents one of the major public health problems in the DRC [11].

The vector responsible for transmission of this parasitic disease is represented by several species of *Simulium* throughout the country, in particular *S. damnosum*, *S. neavei*, and *S. albivirgulatum* [26,27].

The city of Kinshasa, the capital of DRC, is also affected by river blindness. There are three onchocerciasis foci in the capital: the foci of Kinsuka-pêcheurs, Nsele, and Mont-Ngafula. Several authors mentioned the presence of the disease vector in these foci, in particular by the *Simulium* of the *damnosum* complex (*S. squamosum*), but also *S. albivirgulatum* [26,27]. APOC reported that the DRC is a global reservoir of onchocerciasis [28] in which almost 38 million people are at risk [11], of whom approximately 13 million were affected in 2013; the number of people blinded was estimated at 70 thousand [17,28,29,30]. The most recently available data in the 2016 study Global Burden of Disease (GBD) estimated a global prevalence of 14.65 million [2] people, including 12.22 million suffering from the skin disease and 1.03 million cases of sight loss due to onchocerciasis [2]. Onchocerciasis is the third most common cause of blindness in the DRC after cataracts and glaucoma [11]. The DRC has a bite rate of 13 thousand bites per day in the Inga site in Congo Central, the highest black fly bite rate in the world [28].

Onchocerciasis in the DRC is essentially transmitted by two vector groups. On the one hand, black flies belonging to the *Simulium damnosum* complex are the most abundant and are reported alone or in combination with others in almost all foci. On the other hand, *Simulium squamosum*, which is responsible for transmission in the Kinshasa and Congo Central foci, is the only species of this complex which was identified precisely using cytotaxonomic and molecular techniques. Furthermore, two species of the *S. neavei* complex were identified in the DRC, specifically *S. neavei* s.s. in Sankuru and *S. woodi* in Ituri. In addition to these two main groups, the species *S. albivirgulatum* was reported particularly in Kinshasa, Equateur, and in Kwilu in the DRC. It must be noted that data on the entomological situation of onchocerciasis are not well documented and, above all, are not up to date due to the rarity of studies [11] (Table 1).

### 3.6. Socio-Economic Impact

According to a study by the World Bank, prior to the creation of NPOC, onchocerciasis already caused the DRC to lose more than 500 thousand years of productive working life [12]. In the same study, the World Bank mentioned that four million exposed persons in the DRC, i.e., more than 15.4% of people, are carriers of one or more symptoms and complications of the disease [12].

By way of illustration, the following elements can be highlighted: (1)The decrease in production represented by 2000 disability-adjusted life years (DALYs) for 100,000 people, i.e., 562,396.24 DALYs for the DRC, due to the fact that a person losing their sight due to onchocerciasis loses 12 years of productive working life and 20 years if they die blind [12];(2)The phenomenon of stigmatization: carriers of the disease are often rejected by society [1];(3)The decrease in school attendance: 40% of children whose parents suffer from onchocerciasis do not go to school. This indirect consequence of onchocerciasis is due, on the one hand, to the fact that blind people are generally guided by their school-age children, and, on the other hand, to the fact that parents rendered less productive by this disease are unable to send their children to school [13].

### 3.7. Implementation and Process of CDTI Projects

The DRC covers a total of 22 CDTI projects. The first CDTI project, project TIDC Kasaï, was launched in 2001 in the western and eastern regions of Kasaï province, followed in 2002 by Uele (Orientale province). In 2003, four new projects were set up: Bandundu (Bandundu province), Tshopo (Orientale province), Congo Central/Kinshasa (Kinshasa and Congo Central provinces), and Sankuru (Kasaï-Oriental province). Three projects in Katanga (North Katanga, South Katanga, and Lualaba) and five in Equateur province (Tshuapa, North Ubangi, South Ubangi, Mongala, and Equateur-Kiri) were added in March 2004. Kasongo (Maniema province) followed in 2007. In 2008, the projects Butembo-Beni (North Kivu province), Lubutu (Maniema province), Masisi-Walikale (North Kivu province), Rutshuru Goma (North Kivu province), and North Ituri (Orientale province) were launched. In 2012, the South Ituri project started. In 2016, the last project began in South Kivu (Table 2).

It should be mentioned that the main operations of NPOC concentrated on the mass distribution of ivermectin, for a specific approach in the context of co-endemicity of onchocerciasis and loiasis. Mass treatment with ivermectin is recommended in zones where microfilarodermia prevalence is ≥5%. A total of 266 health zones (zones de santé, ZS), i.e., 51.5% of national territory, are eligible for mass treatment with ivermectin. However, from the perspective of the elimination of onchocerciasis, the refinement and mapping in hypoendemic zones is crucial. Financing for the activities was provided by the Congolese government and its partners, which include WHO, LSTM (Liverpool School of Tropical Medicine), CBM (Christian Blind Mission), UFAR (United Front Against River Blindness), SSI (Sight Savers International), The END Fund (which is the only private philanthropic initiative solely dedicated to ending the most common neglected tropical diseases (NTDs). The END Fund is a tax-exempt charitable organization registered in the United States and a company limited by guarantee registered in England and Wales and a UK registered charity and the US Agency for International Development (USAID) ENVISION Program [11].

### 3.8. CDTI Cycles in DRC

In 1987, thanks to the success of the vector control program and the effectiveness of ivermectin as a single medicine for the mass treatment of onchocerciasis, this created great enthusiasm for the activities of the APOC. The proof of the elimination of onchocerciasis in Mali and Senegal led the APOC to change its paradigms, moving away from the elimination of onchocerciasis as a major public health problem in Africa, with CDTI as a tool, toward the reduction of the infection and transmission of onchocerciasis [32]. This happens when ivermectin is administered every year to all people aged five years and above, for a period of at least 15 years without interruption, with the exception of women who are pregnant or breastfeeding during the first week after birth [32,33,34]. As this treatment only takes place once a year, the elimination of onchocerciasis could only be effective with a therapeutic coverage of 80% or more and a geographic coverage of 80% to 100% over more than 15 years without interruption [33,34,35]. In fact, ivermectin is a microfilaricide which has no direct action on the adult filariae of *Onchocerca volvulus* [32,33,34].

Shortly before the cessation of its activities in 2014, the following results were reported by the APOC in the DRC: ivermectin was distributed to 28,251,053 people, of whom 26,049,139 required treatment for onchocerciasis, which represents an average national coverage of approximately 60% [13]. Since the beginning of CDTI, the number of people treated is rising from year to year, but is yet to reach a stable level of therapeutic coverage of 80% or more (Figure 5).

All the CDTI projects covered 42,778 endemic villages; however, 15,700 villages were not treated (36.7%) [10]. The population was estimated at almost 30 million, of whom 7,681,995 were not treated (25.9%) [10]. On the basis of these results, it is difficult to state that onchocerciasis will be eliminated by 2025, as WHO plans, due inter alia to these untreated villages and people [10]. All the zones not covered by CDTI are, thus, obstacles to the new objective of the APOC, namely the reduction of onchocerciasis infection and transmission by 2025 [10].

Moreover, in spite of the APOC’s 15 years of activity, therapeutic coverage is below 80% and the elimination conditions are due to various factors, according to the given context. By way of illustration, a study carried out in the DRC concluded that one of the challenges for CDTI is the occurrence of serious adverse reactions to ivermectin in hyperendemic zones, where there may or may not have been co-existence of onchocerciasis and loiasis [10].

### 3.9. Monitoring and Evaluation

The follow-up and evaluation of the onchocerciasis control program involves coverage surveys, as well as epidemiological, parasitological, and entomological evaluations. Coverage surveys should be carried out in pre-selected districts to validate the coverage of the program. Epidemiological and parasitological evaluations should also be carried out after 10 treatment cycles and be supplemented by entomological surveys, which are yet to take place. To understand the progress of DRC toward the elimination of onchocerciasis, impact assessments must be carried out in various foci in the country where onchocerciasis is endemic.

Moreover, there are 266 health districts (ZS) with endemic onchocerciasis and 253 districts (ZS) with unknown transmission in DRC. We report the presence of 22 foci of onchocerciasis infection with close to 38 million people at risk of contracting onchocerciasis and becoming blind as a result [11].

Shortly before APOC ceased its activities, the DRC achieved 74.1% and 63.3% average geographic coverage in all the known endemic districts and average therapeutic coverage, respectively [10]. From the beginning of CDTI in 2001 until 2017, the average therapeutic and geographic coverages reached 49% and 64.4% respectively (Figure 5). Impact assessment using epidemiological, parasitological, and entomological evaluations is poorly documented. Some preliminary studies of the impact of onchocerciasis carried out by the NPOC in the DRC showed that some districts in the process of transmission are capable of maintaining endemicity in the absence of CDTI. Entomological evaluations are important for deciding to stop the mass distribution of medication in zones of transmission. In the future, they could be supervised directly by the School of Public Health in Kinshasa.

Fly collection sites exist in various parts of the country, with the aim of confirming or contradicting the interruption of transmission and to monitor transmission across internal and external borders. A study was carried out in 2012 to assess the state of onchocerciasis in Kinshasa after more than 10 years of ivermectin-based annual treatment (CDTI). The result showed that the onchocerciasis transmission level index was zero, probably due to CDTI in the capital [27].

Comparing this result to the initial prevalence measured at the time of the REMO surveys, a growing effort toward reduction of the prevalence of microfilariae and the average intensity was noted, while the number of treatments distributed increased [13].

### 3.10. Cross-Border Collaboration and Partnership

The transmission zones of onchocerciasis appear to cross the borders between the DRC and Congo Brazzaville, Rwanda, and Uganda. In effect, there is continual movement of potentially infected people in these countries. The NPOC began cross-border activities with Congo Brazzaville and started collaboration to determine the existence of cross-border transmission. It also aims to improve the monitoring of this transmission. The DRC also reinforced activity with the eastern part of Uganda. Entomological and epidemiological surveys were started by both countries in order to allow Uganda to decide whether or not to stop treatment in the regions bordering the DRC. It must be noted that the DRC, for its part, is yet to launch these surveys. Efforts to control onchocerciasis are, however, gradually getting under way, but the DRC must continue to treat endemic communities in this border region with Uganda.

From 1988 to 2015, several partners supported the function of the NPOC, among them were the World Bank, WHO, APOC, the Rural Health Program (SANRU), and the Christian Blind Mission International (CBM). Their activities were financed by the Government and numerous partners (WHO, CNTD, CBM, UFAR, SSI, END FUND, and USAID ENVISION Program). This financing served, on the one hand, to determine the distribution of the disease in the DRC by means of the REMO surveys, and, on the other, to support the implementation of activities (planning, education, awareness, advocacy and mobilization, training, counting, institutional support, and community treatment) to support community-directed ivermectin treatment [22]. It must be noted that after 17 years of financing, the problem persists and prevalence is only increasing. In fact, the various annual reports from the National Onchocerciasis Task Force (NOTF) show that therapeutic coverage varies between 50% and 70%, and geographic coverage varies between 60% and 90% [13] (Table 3).

In 2015, after the APOC ceased its activities, many partners stopped funding.

### 3.11. Emerging and Re-Emerging Neglected Tropical Diseases

The emergence and re-emergence of old and new diseases is a major problem. Onchocerciasis is one of 17 neglected tropical diseases (NTDs) reported by WHO [35,36].

All 17 NTDs have a disproportionate impact on the world’s poorest people and represent a significant and underestimated global burden of disease. They also constitute a major obstacle to development efforts aimed at reducing poverty and improving human health [35,36].

NTDs are also classified as emerging or re-emerging infectious diseases that represent an even more serious threat and that are not sufficiently examined or discussed in terms of their unique risk characteristics [35,36].

The emergence and re-emergence are accelerated by rapid human development, including many demographic and environmental changes. In this context, NTDs always lacked attention in international public health efforts, leading to options for preventive and insufficient treatment [35,36].

In addition, by 2050, Hotez estimated that regional conflicts related to transfers and limited resources, particularly water, will lead to the collapse of health system infrastructure and, thus, promote the emergence and re-emergence of diseases [35].

Mackey et al. reported that the incidence of some NTDs such as lymphatic filariasis, onchocerciasis, schistosomiasis, and soil-borne helminthiasis would be significantly reduced if preventive chemotherapy against these NTDs (also administered as part of mass treatment) were to be generalized in countries where these diseases are more widespread. In order to prevent the re-emergence of both onchocerciasis and other NTDs, the DRC can only spread or generalize CDTI throughout the country, with onchocerciasis being endemic in all provinces [36].

The DRC should, therefore, study in depth the problem of the emergence and re-emergence of infectious diseases in order to put in place, as of now, adequate prevention strategies to avoid possible reappearances of the disease after elimination in endemic areas.

## 4. Recommendations

### 4.1. Epidemiological and Entomological Monitoring and the Effective Implementation of the Onchocerciasis Control Program in DRC

We recommend the following activities for effective implementation of the onchocerciasis control program in the DRC:Carry out epidemiological and entomological assessments on the country as a whole to better orientate impact assessments in the context of eliminating the disease.Bring together the country’s entomologists for field work and intensify collaboration with neighboring countries affected by onchocerciasis.Update the mapping of onchocerciasis and define unknown transmission zones.Assess the level of endemicity of loiasis in the health zones where this is not currently known, to be mapped using serological testing.

### 4.2. Management and Support of the Process of Verifying the Elimination of the Disease

A focal point within the program should be designated for implementing activities to eliminate onchocerciasis, which will work with other partners including the country’s universities.

### 4.3. Strategies for Eliminating Onchocerciasis

We recommend biannual treatment in certain foci after situational analysis. Awareness among the population should be raised to achieve mass participation and appropriation of CDTI. The serious adverse reactions of ivermectin should be managed optimally. The establishment of a laboratory should be accelerated, allowing transmission of the disease to be evaluated on the basis of polymerase chain reaction (PCR), and reinforcing the training of all staff involved in the CDTI process.

Foci where vector control can be established should be identified with a view to accelerating elimination or reducing the problem (Table 4).

## 5. Difficulties and Challenges

Although many efforts for onchocerciasis control exist in the DRC and are even to be encouraged, there are still some difficulties and challenges to be revealed.

Regarding the difficulties, firstly, the DRC is taking a long time to finalize the mapping of onchocerciasis allowing CDTI. This difficulty is, among other things, linked to the insecurity that occurs in the country, and especially in the east. Secondly, there appears to be a lack of financial resources to achieve the appropriate mapping for the country. Thirdly, the conduct of epidemiological assessments is not yet assured. Finally, there was a delay in completing the mapping of areas of unknown status and in the WHO guidelines for conducting epidemiological assessments. In addition, it is also worth noting the low therapeutic coverage in households that experienced the occurrence of serious side effects (Table 5).

Regarding the challenges, it would be very useful for the DRC to complete the mapping of areas of unknown status or transmission area, to conduct assessments in households with more than 10 treatment cycles and to carry out elimination mapping (Table 5).

In others words, the challenges are as follows: absence of new directives for the assessment of the impact of treatment of onchocerciasis and the mapping of unknown transmission zones in the DRC; absence of epidemiological data; low expertise in the entomology of onchocerciasis; late starting of Computed Tomography (CT) Scan in the health zones which will be eligible following the mapping of unknown transmission zones (Table 5).

## 6. Conclusions

The DRC planned to interrupt the transmission of onchocerciasis by 2020 and to be certified free of the disease by 2025. This objective requires the implementation and rigorous monitoring of the program in collaboration with the development partners. Complete geographic coverage and optimal therapeutic coverage must be recommended. To ensure maintenance of a high coverage of CDTI, surveys of the CDTI coverage must be carried out regularly. The program must aim to improve the active participation of communities in selecting community-directed distributors of medicines, determining CDTI sites and providing appropriate incentives (monetary and non-monetary) to community-directed distributors (CDD). Health workers and CDDs must make concerted efforts to improve knowledge about the disease in the communities, adherence to CDTI, and attitudes to the program in endemic districts, in order to increase CDTI coverage and interrupt the transmission of onchocerciasis. Furthermore, it is essential to improve capacities at all levels to ensure good management of the program. The partnership between all the parties involved must be reinforced to facilitate open discussions concerning the program and, thus, to allow the transfer of knowledge in order to accelerate the control and elimination of the disease. The DRC must plan other feasible interventions, such as vector control and/or the elimination of vectors, and, where necessary, also change the frequency of treatment from annual to bi-annual. Particular attention must also be paid to the training, awareness, and education of CDDs, health workers, and populations, in order to obtain not only their mass participation in CDTI but also their empowerment.

## Figures and Tables

**Figure 1 tropicalmed-04-00092-f001:**
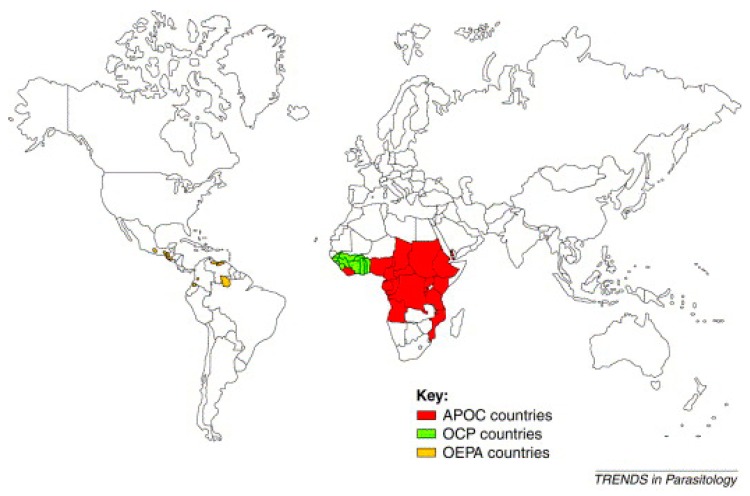
World map of onchocerciasis control programs [21].

**Figure 2 tropicalmed-04-00092-f002:**
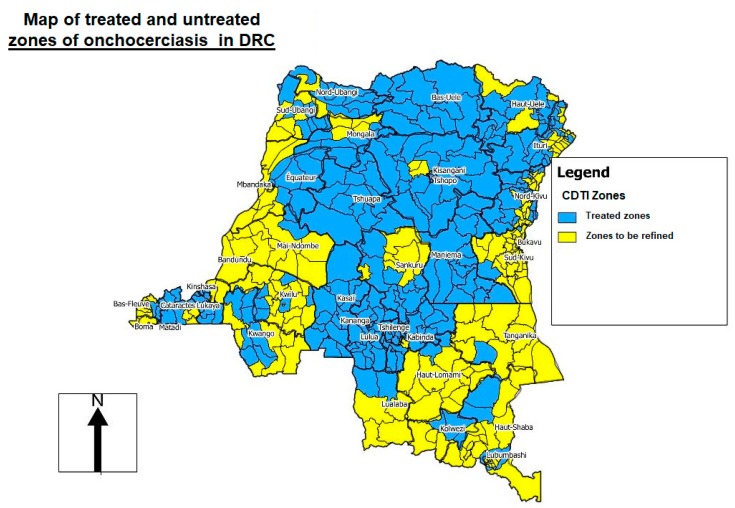
Rapid epidemiological mapping of onchocerciasis (REMO) in the Democratic Republic of the Congo (DRC), showing areas (in blue) where community-directed treatment with ivermectin (CDTI) is needed 2019 [22].

**Figure 3 tropicalmed-04-00092-f003:**
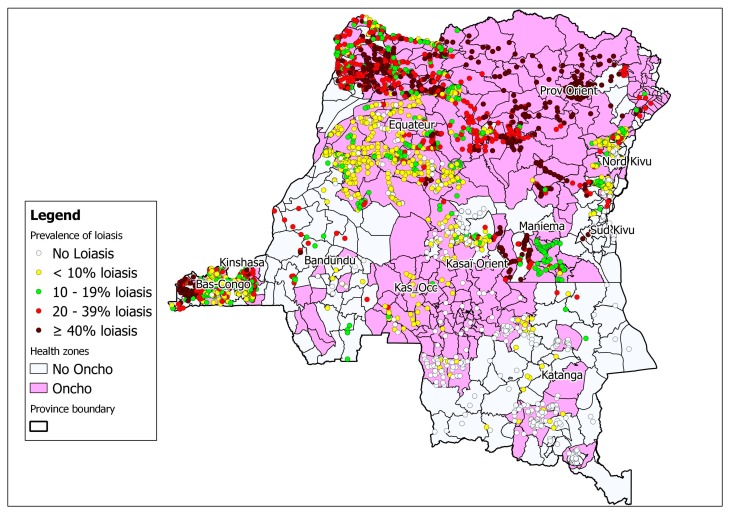
Map of the DRC showing areas co-endemic for onchocerciasis and loiasis (rapid assessment procedures for loiasis and onchocerciasis rapid epidemiological assessment (RAPLOA-REA)) 2012 [17].

**Figure 4 tropicalmed-04-00092-f004:**
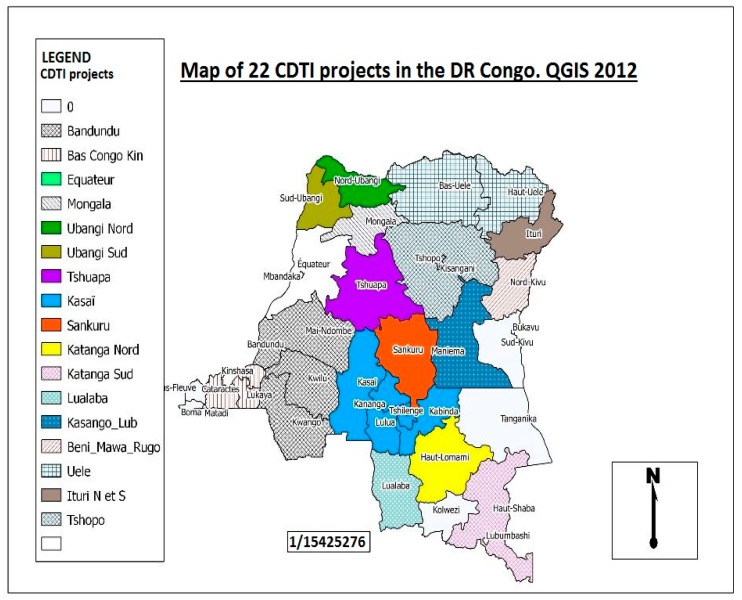
Rapid epidemiological mapping of onchocerciasis in the Democratic Republic of Congo, showing all 22 CDTI projects 2012 [13]. The map shows that onchocerciasis is present in all provinces of the country. Other CDTI projects not mentioned in the legend but visible on the map are: Ituri health zone which have 2 projects, Lubutu, Masisi Walikale, Mongala and Rutshuru Goma.

**Figure 5 tropicalmed-04-00092-f005:**
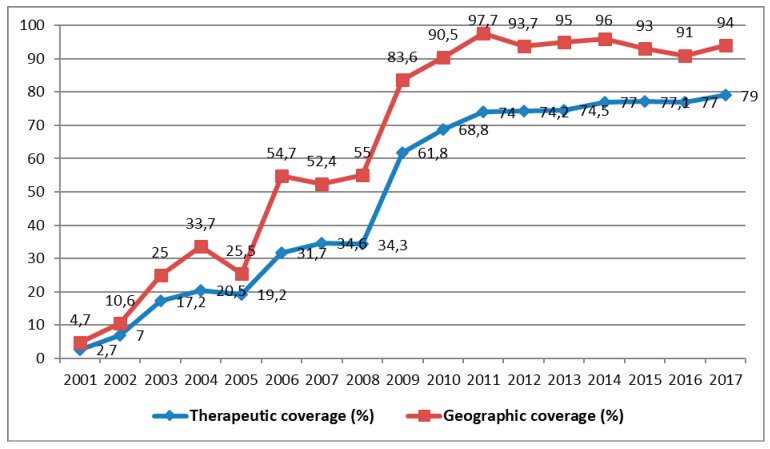
Evolution of therapeutic and geographic coverage of treatment for onchocerciasis from 2001 to 2017 (Source: MS/SG/PNMTN-CTP 2018).

**Table 1 tropicalmed-04-00092-t001:** Distribution of black flies, vectors of *Onchocerca volvulus*, by continent and environment [31].

**Distribution**	**Species**
Africa	*Similium damnosum*
	*Similium neavei*
	*Similium albivirgulatum*
Americas	*Similium ochraceum*
	*Similium metallicum*
**Distribution**	**Subspecies**
Savannah	*S. damnosum* s.s. + *S. sirbanum*
Wetlands of West and Central Africa	*S. sanctipauli* + *S. soubrense*
Forest and uplands of West and Central Africa	*S. squamosum* s.s. + *S. yahense*
Cameroon Mountains of East Africa	*S. mengense* *S. kilibanum*

Onchocerciasis in the Democratic Republic of the Congo (DRC) is essentially transmitted by two vector groups: *Simulium damnosum* and *Simulium squamosum*, but *S. neavei* and *S. albivirgulatum* were also identified in the DRC.

**Table 2 tropicalmed-04-00092-t002:** Community-directed treatment with ivermectin (CDTI) projects in the DRC, year of creation, and projects co-endemic with loiasis.

	CDTI Projects	Year of Creation	Co-Endemicity of Onchocerciasis and Loiasis
1	Kasaï	2001	No
2	Uele	2002	Yes
3	Bandundu	2003	No
4	Congo-Central/Kinshasa	2003	No
5	Sankuru	2003	Yes
6	Tshopo	2003	Yes
7	Tshuapa	2004	Yes
8	Ubangi Nord	2004	Yes
9	Ubangi Sud	2004	Yes
10	Mongala	2004	Yes
11	Katanga Nord	2004	No
12	Katanga Sud	2004	No
13	Lualaba	2004	No
14	Equateur Kiri	2004	Yes
15	Kasongo	2007	Yes
16	Rutshuru Goma	2008	No
17	Lubutu	2008	Yes
18	Masisi Walikale	2008	Yes
19	Ituri Nord	2008	Yes
20	Beni Butembo	2008	Yes
21	Ituri Sud	2012	Yes
22	Sud Kivu	2016	Yes

The co-endemicity of onchocerciasis and loiasis concerns 15 CDTI projects out of 22 implemented in the DRC.

**Table 3 tropicalmed-04-00092-t003:** Current situation of the National Program for Onchocerciasis Control (NPOC) partners at the national and international level in the DRC.

COORDINATION	PARTNER
1 Katanga Nord	LSTM, SCI, RTI
2 Ubangi Nord	LSTM, SCI, ESPN,
3 Ubangi Sud	END FUND/CBM
4 Equateur	END FUND/CBM
5 Tshuapa	END FUND/CBM
6 Mongala	END FUND/CBM
7 Tshopo	LSTM, SCI
8 Ituri Sud	ESPN, WHO
9 Masisi Walikale	ENDFUND/CBM
10 Bas Uele	SCI, LSTM
11 Kasai Kananga	LSTM, END FUND/CBM

SCI: Schistosomia Control Initiative; RTI: research triangle institute; ESPN: expanded special project for elimination of neglited troipical deseases.

**Table 4 tropicalmed-04-00092-t004:** Activities encouraging long-term compliance with CDTI in the DRC.

Objectives	Specific Activities	Targeted Projects
1. To promote integration of CDTI into other health care services	Planning workshop for the implementation of CDTI in the coordination of non co-endemic projects	All non co-endemic projects
2. To support strong partnership	To raise awareness among the various partners	All CDTI projects
3. To maintain high rates of therapeutic (>65%) and geographic (100%) coverage	- Support the drawing up and implementation of a plan for managing serious adverse reactions (SAR) by the projects - Support coordination in advocacy and awareness raising among the various political and community leaders concerning CDTI - Support the organization of post-distribution campaign coverage surveys	All projects co-endemic with loiasis
4. To promote strong community empowerment	To intensify awareness raising in the community and advocacy with community leaders.	All CDTI projects
5. To promote strong governmental engagement	To lead action with the government of the DRC for financial support for the projects	All CDTI projects
6. To set up a strong information, education, and communication (IEC) strategy, which encourages continuous treatment	To organize engagement and awareness raising sessions with communities	All CDTI projects

**Table 5 tropicalmed-04-00092-t005:** Summary of the strengths, weaknesses, opportunities, and threats of CDTI implementation in the DRC.

No.	Intervention	Strengths	Weaknesses	Constraints	Opportunities
1	Management of pilot organizations	Involvement of the Secretary General of Health	- Insufficient meetings of the GTNO - Poor monitoring of recommendations	Insufficient financing for the national onchocerciasis task forces (NOTFs)	Presence of Non-governmental development organizations (NGDOs) for onchocerciasis control
2	Monitoring and assessment	- Updating directives for the implementation of CDTI in co-endemic zones with high prevalence of Loa Loa - Categorization of CDTI coordination by performance achieved	- Inadequate supervision of CDTI projects - Inadequate feedback after monitoring carried out in CDTI coordination - Noncompliance with the schedule for monitoring and assessing CDTI projects (Independent participatory monitoring, sustainability)	Insufficient funds allocated for monitoring and assessment activities	Presence of partners committed to combating onchocerciasis and other neglected tropical diseases (NTDs) by preventive chemotherapy (PC) interventions
3	Development of human resources for health	The creation of a pool of enhanced skills for the program	Instability and demotivation of trained personnel	Inadequacy of a human resources development policy	Application of specific Human Resources for Public Health Ministry statutes
4	Support for the drug industry	Continuous availability of Mectizan in CDTI coordinations	Non-alignment of Mectizan supply in National Essential Drug Supply System	Taxes and various fees	Presence of a Mectizan Donation Program (MDP) and other donors
5	Support for health zones in public health interventions	249 health zones organized mass distribution campaigns for Mectizan No deaths recorded following this distribution	- Inadequate financial resources of partners - Lack of financial contribution from central government - Existence of cases of refusal/absence from treatment	- Low level of implementation of the budget allocated to MSP - Presence of co-endemic zones with high prevalence of Loa Loa	Presence of partners committed to combating onchocerciasis - The reform of Management Support Unit of the Ministry of Health financing - Organization of integrated campaigns

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
