# Peer review of "Review of the National Program for Onchocerciasis Control in the Democratic Republic of the Congo"

_tropicalmed, 2019, doi:10.3390/tropicalmed4020092_

Reviewer 1 Report

This paper describes the history and achievements of onchocerciasis control and elimination efforts in DRC. This information is crucial for refining estimates of the required remaining duration of interventions for achieving elimination, and is therefore valuable for planning of future interventions. It is also gives insights into the many factors that drive the success of such programmes. I have some questions, comments and suggestions for further improvement: MAJOR COMMENTS: 1. Lines 209-2015: is it possible to add one or more maps, showing 1) baseline onchocerciasis endemicity (based on REMO and/or REA), 2) baseline RAPLOA prevalence, 3) CDTI projects areas that have been treated until now possibly also showing the number of treatment rounds per area, 4) low-endemic areas still untreated, 5) areas where further mapping is needed? I think that this would be really valuable for readers. Some maps are available on the ESPEN website. Are the maps on the ESPEN website fully consistent with your information? 2. The structure of the manuscript can be improved, in particular in section 3. It might be useful to have a separate sub-section to describe international control programmes. The section can include the information on large-scale international control programmes (now included in section 3.1) and control strategies employed internationally (now included in section 3.2). The current sections 3.1 and 3.2 can than focus solely on DRC. 3. Please elaborate a bit more in section 5 on difficulties and challenges. MINOR COMMENTS 4. Lines 197-199 and table 2: I guess that APOC’s initially only targeted the meso- and hyper-endemic zones. Has the definition of the treatment areas been adjusted when APOCs target shifted from morbidity control to true elimination of transmission? Am I correct to assume that the 22 CDTI projects in table 2, only include the previously identified meso- and hyper endemic areas, and not the lower-endemic areas that potentially still need treatment? Clarify in the text. 5. Lines 49-51, 57-58:  the authors provide estimates of the population at risk, number of people infected or affected by clinical manifestations, Burden of disease in DALYs. As such number are likely to have change considerably over time thanks to interventions, please make clear to what calendar year the numbers are applicable. Check remainder of the text for similar issues. 6. Lines 82-86: explain how REA differs from REMO 7. Line 88, 345, figure 2, line 362, table 5, and possibly elsewhere: please define therapeutic and geographic coverage. This is particularly relevant as countries are using different definitions, using either the total population or the estimated eligible population as denominator. If total population is used as denominator, 80% is about the maximum coverage that can be achieved, in view of the fact that children under 5, chronically ill, and pregnant / lactating women are to be excluded. Note that table 5 suggests that 65% therapeutic coverage is sufficient, instead of the 80% mentioned elsewhere. This may be due to such definition problems. 8. Lines 174-175: replace “if necessary” by “where feasible”. Has vector control ever been attempted in DRC? 9. Lines 202-208: does the PNLMTN-CTP only cover onchocerciasis? It does not make sense to contrast onchocerciasis intervention strategies with a longer list of interventions considered for a wide range of NTDs 10. Table 1: not very informative, consider deleting it. If you prefer to keep it, please consider the following: It seems to me that the PNLO of DRC was – in some way - part of APOC. Can that somehow be captured in the table? The end year of APOC should be added. 11. Figure 1, legend: please refer to the primary source of this picture (apparently a paper in Trends in Parasitology) 12. Line 240-244: where does the idea come from that geographic coverage

Author Response

REVIEWER 1

This paper describes the history and achievements of onchocerciasis control and elimination efforts in DRC. This information is crucial for refining estimates of the required remaining duration of interventions for achieving elimination, and is therefore valuable for planning of future interventions. It is also gives insights into the many factors that drive the success of such programs. I have some questions, comments and suggestions for further improvement:

MAJOR COMMENTS:

1. Lines 209-2015: is it possible to add one or more maps, showing 1) baseline onchocerciasis endemicity (based on REMO and/or REA), 2) baseline RAPLOA prevalence, 3) CDTI projects areas that have been treated until now possibly also showing the number of treatment rounds per area, 4) low-endemic areas still untreated, 5) areas where further mapping is needed? I think that this would be really valuable for readers. Some maps are available on the ESPEN website. Are the maps on the ESPEN website fully consistent with your information?

Answer: The National Program for Onchocerciasis Control (NPOC) has several maps concerning onchocerciasis and to be more informative for readers, we have chosen to add the following maps:

a) Map of DRC showing the areas of Community-Directed Treatment with Ivermectin (CDTI Projects).

b) Map of DRC showing areas co-endemic for onchocerciasis and loiasis (RAPLOA-REA).

c) Rapid epidemiological mapping of onchocerciasis in Democratic Republic of Congo, showing meso-/hyper endemic areas (in orange/red/brown) (REMO).

2. The structure of the manuscript can be improved, in particular in section 3. It might be useful to have a separate sub-section to describe international control programs. The section can include the information on large-scale international control programs (now included in section 3.1) and control strategies employed internationally (now included in section 3.2). The current sections 3.1 and 3.2 can than focus solely on DRC.

Answer: For a better understanding for readers, we have restructured the paragraphs in section 3 as follows:

3.1. History of the programs for onchocerciasis control in the world.

3.2. History of the National Program for Onchocerciasis Control (NPOC) in DRC.

3.3. Success of NPOC in endemic areas other than DRC.

3.4. Global control strategy

3. Please elaborate a bit more in section 5 on difficulties and challenges.

Answer: The following comments have been added to deepen section 5 on difficulties and challenges

Although many efforts for onchocerciasis control exist in DRC and are even to be encouraged, there are still some difficulties and challenges to be revealed.

Regarding the difficulties, first, DRC is taking a long time to finalize the mapping of onchocerciasis allowing CDTI. This difficulty is, among other things, linked to the insecurity that has occurred in the country, and especially in the East. Secondly, there appears to be a lack of financial resources to achieve the appropriate mapping for the country. Thirdly, the conduct of epidemiological assessments is not yet assured. Finally, there was a delay in completing the mapping of areas of unknown status and in the WHO guidelines for conducting epidemiological assessments. In addition, it is also worth noting the low therapeutic coverage in households that have experienced the occurrence of Serious Side Effects.

Regarding the challenges, it would be very useful for DRC to complete the mapping of areas of unknown status or transmission area; to conduct assessments in households with more than 10 treatment cycles and to carry out elimination mapping.

MINOR COMMENTS:

 4 Lines 197-199 and table 2: I guess that APOC’s initially only targeted the meso- and hyper-endemic zones. Has the definition of the treatment areas been adjusted when APOCs target shifted from morbidity control to true elimination of transmission? Am I correct to assume that the 22 CDTI projects in table 2, only include the previously identified meso- and hyper endemic areas, and not the lower-endemic areas that potentially still need treatment? Clarify in the text.

Answer: The definition of treatment areas has only changed for hypo-endemic areas. These areas are now called transmission areas or unknown status areas.

Before the transition from control to elimination of onchocerciasis, DRC included only in treatment projects (hotbed) with a nodular prevalence of 20 mf or more. All HZ with a nodular prevalence of less than 20 were considered hypo-endemic and therefore not eligible for treatment.

With the objective of eliminating onchocerciasis, WHO has currently recommended that elimination mapping be redone in areas of unknown status (i.e. hypo-endemic areas at the time) and in non-endemic areas. Therefore, all endemic areas must be treated.

5. Lines 49-51, 57-58: the authors provide estimates of the population at risk, number of people infected or affected by clinical manifestations, Burden of disease in DALYs. As such number are likely to have change considerably over time thanks to interventions, please make clear to what calendar year the numbers are applicable. Check remainder of the text for similar issues.

Answer: we added the years and corrected the text as follows:

World-wide, since 2017, it is estimated that there are 18 million people are infected and present dermal microfilaria; 99% of these people are in Africa. The Global Burden of Disease Study estimated in 2017 that there were 20.9 million prevalent O. volvulus infections worldwide: 14.6 million of the infected people had skin disease and 1.15 million had vision loss

Since 2016, 38 million people, some 41% of the Congolese population, are believed to be at risk of contracting onchocerciasis, and 65 thousand people (1‰ of the population) suffer from blindness.

6. Lines 82-86: explain how REA differs from REMO

Answer: REMO is a rapid assessment of the onchocerciasis's endemicity by nodule detection.  In the DRC, at least 30 villages are selected, including 30 to 50 people aged 20 years old and over per village - who have lived more than 10 years in the village - the distance of 30-50 km between 2 villages along the main river.

 If ≥ 20% of adults have nodules, mass treatment is required and this figure is extrapolated to the entire area. In communities where the nodule rate is less than 20%, clinical treatment is applied. REA is a technique that allows to collect, quickly and at a lower cost, various informations related to the symptoms of onchocerciasis. It's a revised form of REMO.

7. Line 88, 345, figure 2, line 362, table 5, and possibly elsewhere: please define therapeutic and geographic coverage. This is particularly relevant as countries are using different definitions, using either the total population or the estimated eligible population as denominator. If total population is used as denominator, 80% is about the maximum coverage that can be achieved, in view of the fact that children under 5, chronically ill, and pregnant / lactating women are to be excluded. Note that table 5 suggests that 65% therapeutic coverage is sufficient, instead of the 80% mentioned elsewhere. This may be due to such definition problems.

Answer:  Therapeutic Coverage is the number of people treated multiplied by 100 and divided by the total population (exposed population). It must be equal to or greater than 84% each year in each community. Geographical Coverage, on the other hand, is the number of communities treated multiplied by 100 and divided by the total number of communities. It must reach 100% each year.

Although APOC initially recommended a minimum therapeutic coverage of ≥65% for control of onchocerciasis as a public health problem, when its strategy moved from control towards elimination, this target was elevated to ≥80% with a recommended geographical coverage of 100%

Unfortunately DRC's NOTF did not reach either of these goals between the years of CDTI control from 2001 to 2012.

8. Lines 174-175: replace “if necessary” by “where feasible”. Has vector control ever been attempted in DRC?

Answer: We replace « if necessary » by « where feasible ».

Vector control is currently applied only in one hotbed, North Ituri, where traps are used to catch black flies. It should be noted that with the establishment of the committee of independent experts on onchocerciasis control, this control strategy is a priority to achieve the elimination of onchocerciasis in the DRC.

9. Lines 202-208: does the PNLMTN-CTP only cover onchocerciasis? It does not make sense to contrast onchocerciasis intervention strategies with a longer list of interventions considered for a wide range of NTDs

Answer: There was certainly a mistake in translating the term "relayed" from French into English.

Indeed, NPCNTDs-PC does not only cover onchocerciasis but covers all NTDs present in the DRC. As the NPCNTDs-PC substitutes the NPOC, it has taken over both the objectives of combating NTDs and all NPOC missions.

We have therefore replaced the word "supported by" by "was relayed by" and corrected the sentence for a better understanding for the reader.

10. Table 1: not very informative, consider deleting it. If you prefer to keep it, please consider the following: It seems to me that the PNLO of DRC was – in some way - part of APOC. Can that somehow be captured in the table? The end year of APOC should be added.

Answer: we decided to delete Table 1 and give a brief summary of the different international programs involved in the control of onchocerciasis.

11. Figure 1, legend: please refer to the primary source of this picture (apparently a paper in Trends in Parasitology)

Answer: we have inserted the appropriate reference for Figure 1

12. Line 240-244: where does the idea come from that geographic coverage?

Answer: According to APOC, as mentioned in different literature:  therapeutic coverage of 80% or more and a geographic coverage of 80% to 100% are required for at least 15 to 17 annual cycles to eliminate onchocerciasis.

The idea is to show that the DRC has not achieved the required geographical and therapeutic coverage as recommended by APOC.

Unfortunately, in 15 years of CDTI, the DRC has never reached these recommended guidelines in terms of geographical and therapeutic coverage.

Reviewer 2 Report

The authors have reviewed NPOC Program in DC Congo aimed to eliminate Onchocerciasis. The review is extensive and studies are well documented. Article is sectioned properly and majorly covers all aspects of Onchocerciasis control program in DC Congo.  I have following suggestion on this review article.

In Section 3.3. "Epidemiology.........DR Congo", I would suggest to include a map of DR Congo showing epidemiology of Onchocerciasis.

I understand that follow-up data for the current status of Onchocerciasis in DR Congo post-NPOC Program is not available, my suggestion to authors is to include a small section stating success of NPOC in endemic areas other than DR Congo.

Authors can also include a short paragraph stating the reemergence of Onchocerciasis in countries reported to be free of this parasitic infection. Reemergence of helminth parasitic infection post mass drug administration is a major issue and highlighting such data in this manuscript will definitely strengthen this article. 

Author Response

REVIEWER 2

The authors have reviewed NPOC Program in DC Congo aimed to eliminate Onchocerciasis. The review is extensive, and studies are well documented. Article is sectioned properly and majorly covers all aspects of Onchocerciasis control program in DC Congo.  I have following suggestion on this review article.

In Section 3.3. "Epidemiology.........DR Congo", I would suggest to include a map of DR Congo showing epidemiology of Onchocerciasis.

Answer:  we include the Rapid epidemiological mapping of onchocerciasis in DR Congo, showing meso-/hyper endemic areas (in orange/red/brown).

I understand that follow-up data for the current status of Onchocerciasis in DR Congo post-NPOC Program is not available, my suggestion to authors is to include a small section stating success of NPOC in endemic areas other than DR Congo.

Answer: After having carefully implemented the control strategies recommended by APOC, onchocerciasis control program in several countries have been successful, going from control to complete elimination of the disease: Colombia in 2013, Ecuador in 2014, Mexico in 2015 and Guatemala in 2016.

In addition, towards the end of 2017, other countries, Bolivia, the Republic of Venezuela, Uganda and Sudan, also terminated CDTI and in at least one transmission zone, conducted for 3 years post-treatment surveillance.

In general, the elimination of onchocerciasis is possible after strict application of control strategies as recommended by WHO. Each country has its own particularities in order to adapt these strategies according to the reality of their own situation.

We therefore encourage the NPCNTDs-PC to develop efforts and multiply control strategies to eliminate onchocerciasis. Indeed, the results of NPOC such as the increase in therapeutic and geographical coverage from year to year and the increase in the number of community distributors (CDs) and trained health workers (HWs) are to be encouraged. In the first 12 years of CDTI in the DRC, the country had achieved an average of 1 DC for 262 people and 1 HW for 2368 people while the standards recommended by APOC are: 1 DC for 100 people and 1 HW for 2500 to 5000 people.

Difficulties at the beginning include: armed conflicts (1998 -2003 - 2007), serious side effects (death) and the objectives of therapeutic and geographical coverage not achieved due to geographical inaccessibility. 

Authors can also include a short paragraph stating the reemergence of Onchocerciasis in countries reported to be free of this parasitic infection. Reemergence of helminth parasitic infection post mass drug administration is a major issue and highlighting such data in this manuscript will definitely strengthen this article.

Answer: The emergence and re-emergence of old and new diseases is a major problem. Onchocerciasis is one of 17 neglected tropical diseases (NTDs) reported by WHO.

All 17 NTDs have a disproportionate impact on the world's poorest people and represent a significant and underestimated global burden of disease. They also constitute a major obstacle to development efforts aimed at reducing poverty and improving human health.

NTDs are also classified as emerging or re-emerging infectious diseases that represent an even more serious threat and have not been sufficiently examined or discussed in terms of their unique risk characteristics.

The emergence and re-emergence are accelerated by rapid human development, including many demographic and environmental changes. In this context, NTDs have always lacked attention in international public health efforts, leading to options for preventive and insuffisance treatment.

In addition, by 2050, Hotez estimates that regional conflicts related to transfers and limited resources, particularly water, will lead to the collapse of health system infrastructure and thus promote the emergence and re-emergence of diseases.

Tim K. Mackey et al report that the incidence of some NTDs such as lymphatic filariasis, onchocerciasis, schistosomiasis and soil-borne helminthiasis would be significantly reduced if preventive chemotherapy against these NTDs (also administered as part of mass treatment) were to be generalized in countries where these diseases are more widespread. In order to prevent the re-emergence of both onchocerciasis and other NTDs, DR Congo can only spread or generalize CDTI throughout the country; onchocerciasis being endemic in all provinces.

DR Congo should therefore study in depth the problem of the emergence and re-emergence of infectious diseases in order to put in place, as of now, adequate prevention strategies to avoid possible reappearances of the disease after elimination in endemic areas.

Hotez PJ. Human Parasitology and Parasitic Diseases: Heading Towards 2050. Adv Parasitol. 2018; 100:29-38.

Tim K. Mackey, Bryan A. Liang, Raphael Cuomo, Ryan Hafen, Kimberly C. Brouwer and Daniel E. Lee. Emerging and Reemerging Neglected Tropical Diseases: a Review of Key Characteristics, Risk Factors, and the Policy and Innovation Environment. Clin Microbiol Rev. 2014 Oct; 27(4): 949–979.